# Mineralization Regularities of the Bainiuchang Ag Polymetallic Deposit in Yunnan Province, China

**Fuju Jia** [1][ID]**, Ceting Yang** [1][ID]**, Guolong Zheng** [2]**, Mingrong Xiang** [2]**, Xuelong Liu** [1,*]**, Wei Duan** [2]**, Junshan Dao** [2] **and Zhihong Su** [1]

1   College of Land Resources and Engineering, Kunming University of Science and Technology, Kunming 650093, China; jiafujv@kust.edu.cn (F.J.); yangceting@stu.kust.edu.cn (C.Y.)
2   No. 306 Geological Party of Yunnan Nonferrous Geological Bureau, Kunming 650217, China
*   Correspondence: liuxuelong@kust.edu.cn

**Abstract:** The Bainiuchang Ag polymetallic deposit is located at the junction between the Cathaysia, Yangtze, China and Indosinian blocks. It has experienced many geological events, and records excellent conditions for multiple mineralization. In this paper, elemental correlation analysis, cluster analysis, factor analysis, a semivariogram of Zn/Pb values, mineralization distribution and trend surface analysis have been carried out based on the prospecting database and ore body model. Our results show that Ag–Pb–Zn were mineralized at moderate temperatures. Tin was mineralized at high temperatures, and Sn and Zn/Pb values are well correlated. The Zn/Pb values can be used for tracing the ore-forming fluid. The semivariogram revealed that the Zn/Pb values are moderately spatially dependent, with good mineralization continuity in the 100° and 10° directions. The spatial pattern of the elemental grade correlates with mineralization enrichment. The trend surface analysis shows that the Ag, Pb, Zn, and Cu mineralization is weak in the south and strong in the north of the deposit, and the Sn grades and Zn/Pb values are high in the south and low in the north. High-temperature Sn, medium-temperature Cu, and medium-temperature Ag–Pb–Zn mineralization have occurred in a south-to-north trend. Therefore, the source of the ore-forming fluid was in the southern part of the mining area. During the migration of the ore-forming fluid from south to north, different minerals were precipitated due to changes in the physicochemical environment. The spatial patterns of mineralization may provide a basis for studying the formation of the ore deposit, and can guide ore exploration and mining in the mine area and similar ore deposits elsewhere.

**Keywords:** geostatistics; trend surface analysis; Zn/Pb ratio; mineralization regularity





## 1. Introduction

After years of geological exploration and resource development, metallic mines have a significant amount of exploration data. Combining the massive amount of sample analysis data with the 3D spatial attributes adds to the richness and diversity of the geological information. With the help of statistical principles and methods, these data can be studied in depth to summarize the laws of elemental assemblages and spatial mineralization, which are not only the guidance for analyzing the genesis of deposits and searching for ore at depth and at the edges of deposits [1–5], but also can be used as a reference for the formulation of mining plans and the selection of beneficiation methods for deposits.

The Bainiuchang Ag polymetallic deposit in Yunnan Province, China, is located at the junction of the Cathaysia, Yangtze, China and Indosinian blocks, and was affected by the Caledonian, Indosinian and Yanshanian orogenies. This deposit has cumulative proven resources of 6470 tons of Ag, 1,721,400 tons of Zn and 1,096,700 tons of Pb, with a threshold concentration of Ag = 40 ppm, Zn = 0.5 wt.% and Pb = 0.3 wt.%, and is also associated with abundant amounts of Cu and Sn, as well as In, Cd, Ga and Ge [6,7]. In terms of the global Ag output, about two-thirds of the Ag resources are associated with non-ferrous

and precious metal deposits, such as Cu, Pb, Zn and Au, and about one-third are Ag-only deposits. In base metal sulphide ores, Ag is most commonly found in galena, followed by chalcopyrite and then sphalerite [8,9].

The origins of the metallogenic materials and ore-forming hydrothermal fluid in the Bainiuchang deposit are controversial, possibly including a magmatic–hydrothermal origin [10–16], submarine exhalative sedimentation or the combination of both [17–19]. The main bases of the granite-hydrothermal genesis are: (1) the deposit is located in the vicinity of late Yanshanian granite, and the age of mineralization is consistent with the age of granite formation; (2) the rare earth elements, trace elements and isotopic composition of the ore have the characteristics of magmatic–hydrothermal origin; and (3) the output of the ore body is tectonically controlled, with clear boundaries with the surrounding rocks, and has the characteristics of hydrothermal infill mineralization. The main bases of the submarine-exhalative-sedimentary genesis are: (1) the ore-bearing strata have high background values of ore-forming elements and contain pyroclastic debris in the ore-bearing tuffs; (2) the ore body is stratified, developing laminated and banded ore; and (3) the rare earth elements, trace elements and isotopic composition of the ore are similar to other exhalative sedimentary deposits. The Bozhushan granite, located in southeast of the Bainiuchang deposit, is a two-phase composite pluton. The main rock type of the first phase is a biotite monzogranite intruded at 97–104 Ma, while the second phase is mainly fine-grained monzogranite intruded at 48–79 Ma [20]. The mean Rb–Sr isotope age of the hidden granite pluton in the Bainiuchang mining area is $68.8 \pm 2.6$ Ma [6], which is consistent with the age of the Bozhushan granite (phase 2). A cassiterite U–Pb isochron age of $87.0 \pm 3.0$ Ma [12] for the Bainiuchang mining area is broadly consistent with the ages of Yanshanian granites in the Bozhushan area, suggesting that cassiterite mineralization was closely related to these granites.

In this study, we collected ore-grade data of the Bainiuchang deposit and constructed a database based on major and trace elements data acquired over many years of exploration and mining, and conducted the 3D Spatial Modelling of Ag, Pb, Zn, Sn and Cu assay data. This study obtained the combination regularity among ore-forming elements through correlation analysis, cluster analysis and factor analysis, and summarized the spatial mineralization regularity of each element through semivariogram analysis, mineralization contour plotting and trend surface analysis. The source and transport direction of ore-forming fluid were analyzed to explore the genesis of deposits and provide a basis for mineralization prediction and resource development.

## 2. Geological Background

The Bainiuchang deposit is located in the southeast Yunnan metallogenic belt of the Cathaysian block, adjacent to the Yangtze block in the west and the Indosinian block in the south, at the junction of three tectonic units. It has experienced three stages of Caledonian, Indosinian and Yanshanian orogenesis, and the geological processes in the area are complex. Yanshanian granites formed around the periphery of the Youjiang rift basin around a number of large Sn polymetallic deposits, such as at Gejiu and Dachang (Figure 1a).

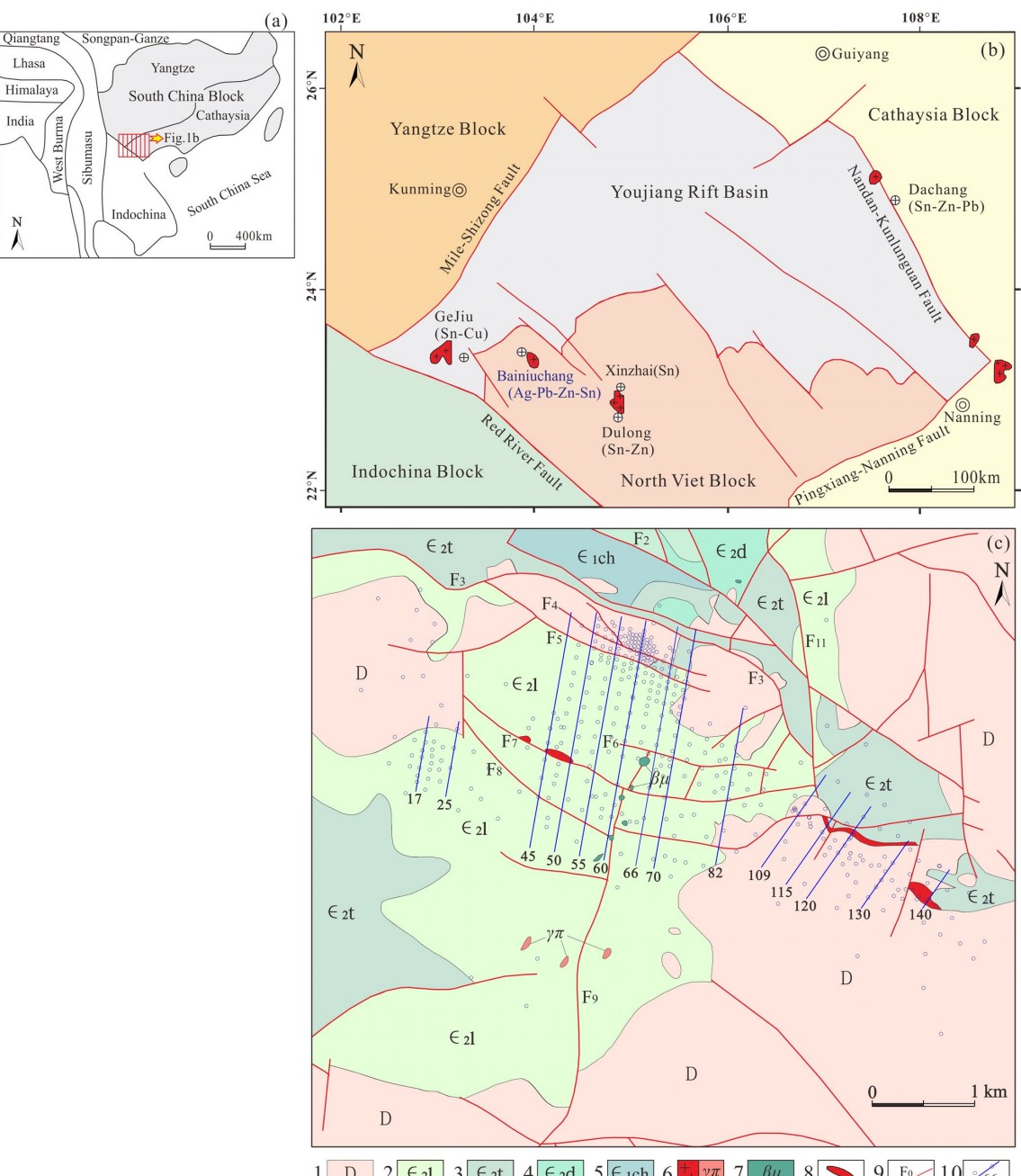

**Figure 1.** Geological maps of the Bainiuchang deposit. (**a**) Simplified map of Eastern Asia showing major tectonic units (modified using Ref. [21]). (**b**) Regional geological map (modified using Ref. [22]). (**c**) Geological map of the Bainiuchang mine area: 1. Devonian strata; 2. middle Cambrian Longha Formation; 3. middle Cambrian Tianpeng Formation; 4. middle Cambrian Dayakou Formation; 5. lower Cambrian Chongzhuang Formation; 6. Yanshanian granite and granitic porphyry ($\gamma\pi$); 7. diabase; 8. ore body; 9. fault; and 10. exploration line and drill hole.

The strata in the Bainiuchang deposit are mainly Cambrian and Devonian in age. The Cambrian rocks are grey and medium-bedded limestone along with siltstone, argillaceous siltstone and mudstone, which formed in a shallow-marine shelf and coastal environment. The ore-bearing strata is mainly in the middle Cambrian Tianpeng Formation ($\in_2 t$) unit (Figure 1b). The faults in the mining area are NW–SE-trending and include $F_2$, $F_3$, $F_4$, $F_5$, $F_6$, $F_7$ and $F_8$, which are all normal faults (Figure 1c). $F_3$ is the main ore-controlling fault, and the main ore bodies occur in the footwall of the fault. Deep prospecting in the southeast

of the mine area has identified a fine- to coarse-grained biotite monzogranite that yields a zircon U–Pb age of $85.26 \pm 0.54$ Ma [6,22].

There are more than 70 ore bodies in the mining area, and ore body $V_1$ accounts for >90% of the total ore reserves. Ore body $V_1$ strikes 100° and dips 190° to the SW (with an angle of 15–20°), and has a strike length of 4.8 km, depth of dip up to 2.5 km (average = 1.31 km) and maximum thickness of 33.62 m (average = 5.65 m). The ore body occurs in the upper part of the middle Cambrian Tianpeng Formation near the $F_3$ fault (Figure 2). The main ore minerals are sphalerite, galena, Ag-bearing sulphides, chalcopyrite and cassiterite. The gangue minerals are quartz, calcite, dolomite and chlorite. The ore mainly has massive, laminated and vein-like forms. The ore has aggregate, mosaic and diffuse textures (Figure 3).

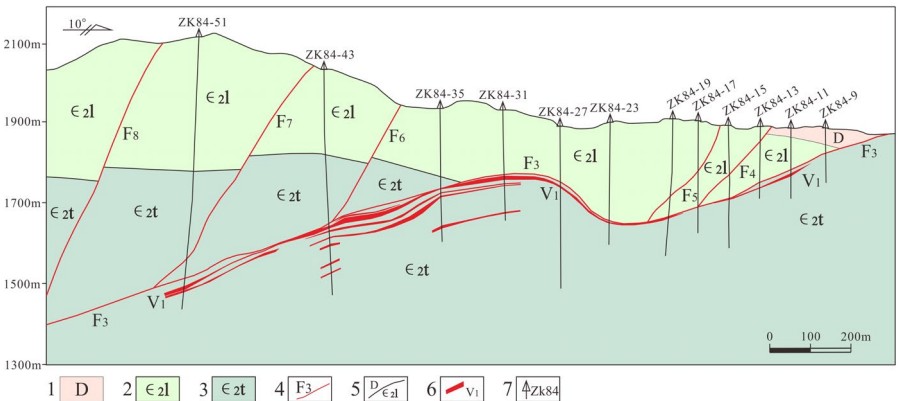

**Figure 2.** Cross-section along exploration line 66 (Figure 1c): 1. Devonian strata; 2. Middle Cambrian Longha Formation; 3. Middle Cambrian Tianpeng Formation; 4. Fault; 5. Geological boundary; 6. Ore body; and 7. drill hole.

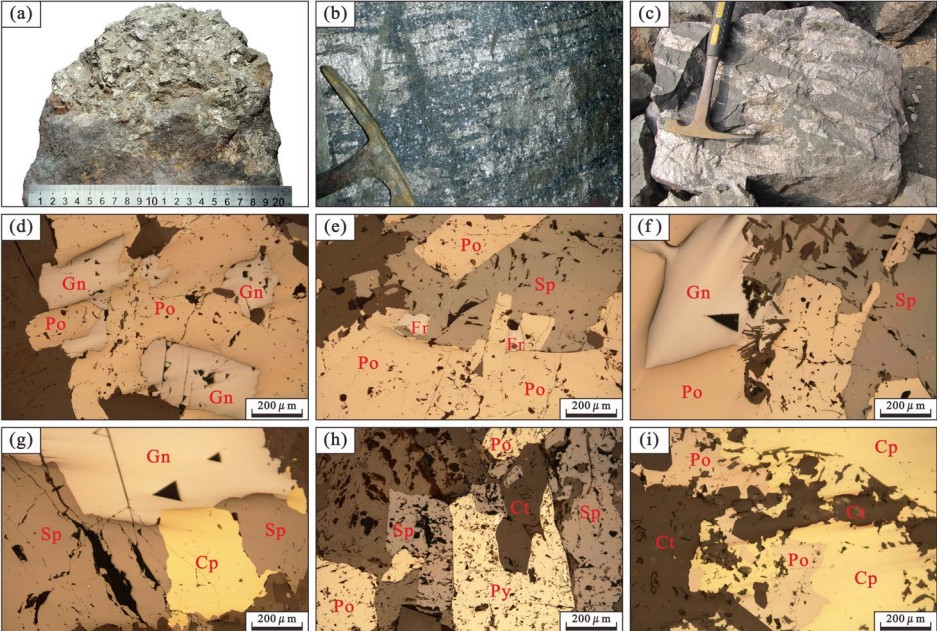

**Figure 3.** Photographs of the ores and their microscopic features. (**a**) Massive ore; (**b**) laminated ore; (**c**) veined ore; (**d**) galena replacing pyrrhotite; (**e**) sphalerite and freibergite replacing pyrrhotite; (**f**) galena and sphalerite replacing pyrrhotite; (**g**) chalcopyrite replacing sphalerite and galena; (**h**) sphalerite and pyrite enclosing automorphic cassiterite, with a mosaic texture; and (**i**) chalcopyrite and pyrrhotite enclosing automorphic cassiterite, with a mosaic texture. Cp = chalcopyrite; Ct = cassiterite; Fr = freibergite; Gn = galena; Po = pyrrhotite; Py = pyrite; and Sp = sphalerite.

According to the characteristics of the main ore minerals, they can be classified as pyrite-pyrrhotite-arsenopyrite-sphalerite ore, sphalerite-galena-marcasite ore, cassiterite ore, galena-sphalerite-pyrrhotite-pyrite ore, pyrrhotite-pyrite-cassiterite-chalcopyrite ore, pyrrhotite-chalcopyrite-pyrite-sphalerite ore, pyrrhotite-chalcopyrite ore, etc.

By the characteristics of the combination of the main ore-forming elements, it can be divided into silver-lead ore, silver-lead-zinc ore, silver-lead-zinc-tin ore, silver-copper-zinc-tin ore, silver-copper ore, lead-zinc ore, lead-zinc-tin ore, and so on. Among them, silver-lead-zinc-tin ore is the most dominant ore, with its reserves accounting for >90% of the total reserve.

## 3. Raw Data and Methodology

We carried out a field geological survey of the Bainiuchang mine area and comprehensively collected and collated data from previous mineral exploration and exploitation. The 3D Spatial Modelling of prospecting results and analytical data was achieved by establishing a geological database, and finally, an ore body model was established. Correlation analysis, cluster analysis and factor analysis were used to explore the correlation and assemblage rule of elements, and the spatial mineralization regularity of elements was analyzed using semivariogram, contours and trend surfaces. An overview of flowchart is presented in Figure 4.

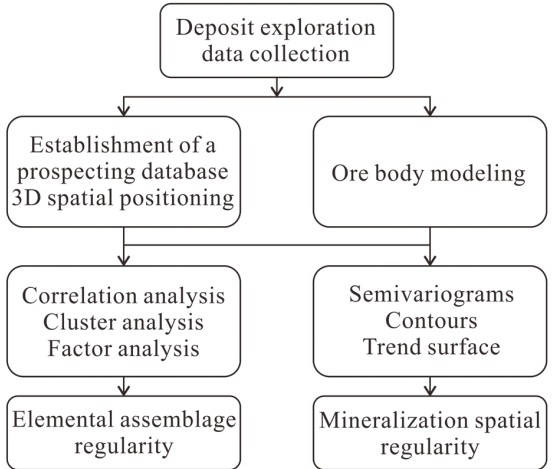

**Figure 4.** The methodological flowchart implemented in this study.

### 3.1. Database and Ore Body Model

The mining area of the Bainiuchang Ag polymetallic deposit is 6.0 km wide from east to west and 4.0 km long from north to south, with a surface elevation of 1712.0–2278.0 m and a maximum drilling depth of 963.8 m. The data required for the construction of the database were mainly derived from the drilling and tunnel catalogues produced during resource exploration and mining, which includes 408 drill holes, 992 tunnel projects and 15,046 samples analyzed for Ag, Pb, Zn, Sn, and Cu (Table 1). After the data were entered into the 3Dmine software, the prospecting results and data were positioned and displayed in three dimensions. An ore body model was established with the cut-off grades of Ag = 40 ppm, Pb = 0.3 wt.%, Zn = 0.5 wt.%, Sn = 0.2 wt.% and Cu = 0.3 wt.% (Figure 5).

**Table 1.** Exploratory operations statistics table.

| Projects | Quantities | Total Length | No. of Samples | Samples of the Ore |
|---|---|---|---|---|
| Drill hole | 408 | 164,631 m | 5967 | 1314 |
| tunnel | 992 | 40,400 m | 9079 | 2776 |
| Total | 1400 | 205,031 m | 15,046 | 4090 |

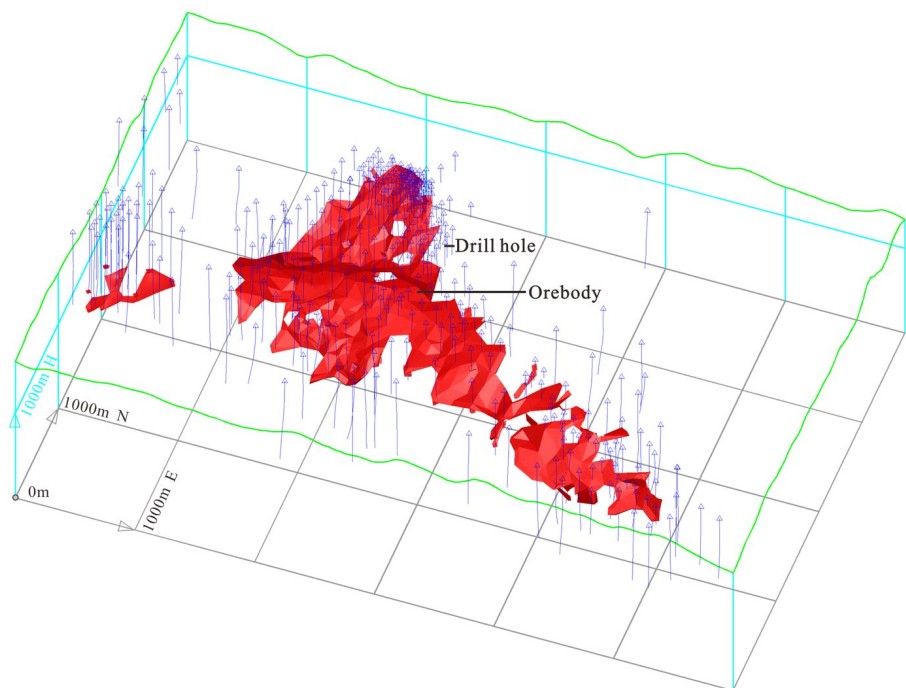

**Figure 5.** Locations of the drill holes and the ore body model.

### 3.2. Correlation and Cluster Analysis

Correlation analysis examines different features and relationships between data to find correlations and identify key factors responsible for the data variations. The correlation coefficient (r) is a statistical parameter of the closeness of the relationship between variables. In this paper, the correlation coefficient was used to examine the correlations between the major metallic elements and to underpin the modelling. The r value varies between 1 and −1, and r > 0 means that two variables are positively correlated, r < 0 means that two variables are negatively correlated and r = 0 means that two variables are uncorrelated. The formula for calculating r is as follows:

$$r_{xy} = \frac{s_{xy}}{s_x s_y} \tag{1}$$

where $r_{xy}$ denotes the sample correlation coefficient, $S_{xy}$ denotes the sample covariance, $S_x$ denotes the sample standard deviation of $x$ and $S_y$ denotes the sample standard deviation of $y$.

Cluster analysis is a statistical technique that classifies research objects (samples or variables) into relatively homogeneous groups or clusters based on their similarity. It is a method for simplifying data via data modelling. In geological studies, R-type cluster analysis is the mathematical study of the degree of similarity in the geochemical behaviour of different ore-forming elements [23,24], and it is one of the most commonly used multivariate statistical methods to study ore paragenesis.

### 3.3. Factor Analysis

As a very useful multivariate statistical analysis technique, factor analysis has been widely used in the interpretation of geochemical data. Factor analysis is a data analysis technique for extracting common factors based on the correlation of variable data sets. It can reduce the dimension of the data set to uncorrelated main components through the covariance or correlation matrix of the variable [25,26], and reveal a more parsimonious data structure of the measured variable [27–29]. In short, it describes the variation of a multivariate data set with as few factors as possible.

The main steps of factor analysis are (1) to perform factor extraction, (2) to calculate the factor scores of the sample and determine the factor variables, (3) to clarify the structure of the variables and (4) to determine whether the data meet the prerequisites for factor analysis.

### 3.4. Geological Significance of Zn/Pb Values

Zn/Pb ratios can be used to trace the source and migration direction of ore-bearing fluid. The geochemical behaviour of Zn and Pb are similar, but exhibit slight differences. Both Zn and Pb commonly occur in nature with a valence of +2. Both are strongly chalcophile and combine with sulphur ions to form sulphides. In terms of atomic structure and crystal chemistry, Zn and Pb are somewhat different, with the atomic and ionic radius of Zn being smaller than Pb. Zn has properties similar to Fe and Mn, while Pb has properties similar to K [30,31].

Given that the physicochemical conditions change during the migration of ore-forming fluid, the different geochemical behaviours of elements and order of crystallisation lead to the formation of specific ores and grades, which result in elemental zoning within mineralization zones. The order of precipitation of ore-forming materials is related to the stability of the metallic elements, with As > Hg > Sb > Ag > Pb > Zn > Cu, and the less stable elements precipitating earlier. Pb/Zn deposits are often sphalerite-rich in their lower parts and galena-rich in their upper parts [32]. Changes in Zn/Pb ratios can be used to trace the source of the ore-forming fluid by plotting Zn/Pb contours, with the center of high values being the source of the ore-forming fluid [33–35].

### 3.5. Semivariogram

Semivariograms describe the structure and spatial variability of data by examining spatial correlations. Semivariogram analysis is a function of distance and direction, and allows the distance and direction in which geochemical data are best correlated or exhibit the best continuity to be identified. The semivariogram equation is as follows [36]:

$$r(h) = \frac{1}{2N(h)} \sum_{i=1}^{N(h)} [Z(x_i) - Z(x_i + h)]^2 \qquad (2)$$

where $r(h)$ is the variance value for a distance $h$, $N(h)$ is the number of data pairs separated by a lag distance $h$, $Z(x_i)$ is the value of property Z in position $x_i$ and $Z(x_i + h)$ is the value of property Z in position $x_i + h$.

The nugget, sill and range are used to describe the nature of the spatial variability in a semivariogram map. The nugget is the semivariogram value at $h = 0$. As h increases, the semivariogram values increase toward the sill, at which point the semivariogram values become uniform. The range is the distance over which the semivariogram values reach the sill.

The nugget represents measurement errors and spatial micro-variability at distances smaller than the sampling interval [37]. The nugget/sill ratio can be used to classify the spatial dependence of variables. Values of ≤0.25 indicate the parameter is strongly spatially dependent; values between 0.25–0.75 indicate the parameter is moderately spatially dependent; values of >0.75 indicate the parameter has a weak spatial dependence [38]. The range represents the maximum distance at which a spatial autocorrelation is present and, if the distance is further than the range, the variables are spatially uncorrelated or independent.

### 3.6. Trend Surface Analysis

Trend surface analysis is commonly used to study regional patterns of change and identify anomalous areas. This method decomposes the changes in variables into trend and residual surfaces [39–41]. The trend surface reflects the regional pattern of variations, which is controlled by systematic factors on a large scale, whereas the residual surface reflects the characteristics of local variations, which are controlled by local and stochastic

factors. The first-order trend surface formula is $f(x,y) = a_0 + a_1x + a_2y$ and the second-order trend surface formula is $f(x,y) = a_0 + a_1x + a_2y + a_3x^2 + a_4xy + a_5y^2$.

The trend surface analysis was undertaken by regression analysis, using all the sample data, and by fitting a first, second, or nth order trend surface and using the difference between the variable and trend surface values to plot the residual surface. The optimal trend surface was selected by validation analysis and comparing the probability value ($p$; When $p < 0.05$, the statistical results of the trend surface are significantly different from the original data, the smaller the better), coefficient of determination ($R^2$; the closer to 1 the better), root-mean-square error (RMSE; the smaller the better), coefficient of variation (CoeffVar; the smaller the better) and the independence of the residuals (Durbin–Watson D; the closer to 2 the better) of the trend analysis results.

## 4. Results

### 4.1. Correlation and Cluster Analysis

The correlation and R-cluster analysis were carried out for Ag, Pb, Zn, Sn, Cu and Zn/Pb values based on 15,046 sets of ore assay data (Table 2 and Figure 6).

**Table 2.** Correlation coefficients.

|  |  | Ag | Pb | Zn | Sn | Cu | Zn/Pb |
|---|---|---|---|---|---|---|---|
| Ag | Pearson Correlation Sig.(2-tailed) | 1.000 | | | | | |
| Pb | Pearson Correlation Sig.(2-tailed) | 0.802 ** 0.000 | 1.000 | | | | |
| Zn | Pearson Correlation Sig.(2-tailed) | 0.563 ** 0.000 | 0.638 ** 0.000 | 1.000 | | | |
| Sn | Pearson Correlation Sig.(2-tailed) | 0.094 ** 0.000 | 0.070 ** 0.000 | 0.167 ** 0.000 | 1.000 | | |
| Cu | Pearson Correlation Sig.(2-tailed) | 0.004 0.782 | −0.127 ** 0.000 | −0.107 ** 0.000 | −0.032 * 0.042 | 1.000 | |
| Zn/Pb | Pearson Correlation Sig.(2-tailed) | −0.095 ** 0.000 | −0.102 ** 0.000 | 0.319 ** 0.000 | 0.110 ** 0.000 | −0.007 0.667 | 1.000 |

*. Correlation is significant at the 0.05 level (2-tailed); **. Correlation is significant at the 0.01 level (2-tailed).

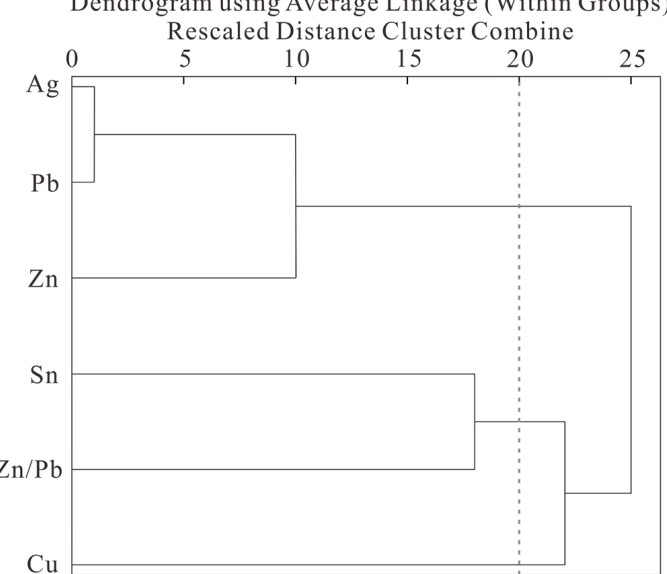

**Figure 6.** R-type cluster analysis results for the ore-forming elements.

The results of the correlation analysis consisted of two components, the Pearson Correlation value (r) and significance. A significant strong positive correlation indicates that the two elements share similar geological processes or the same mineralization space. This study shows that Ag, Pb, and Zn have a significant strong positive correlation. Sn is significant with Ag, Pb, Zn, and Zn/Pb, but the correlation level is mainly weak or uncorrelated. Cu is mainly uncorrelated or negatively correlated with the other elements, suggesting a different mineralization space for Cu and the other elements.

The R-type cluster analysis method was the furthest neighbor and the measuring standard was Pearson correlation. The result shows that the elements can be divided into three groups, which are Ag–Pb–Zn, Sn–Zn/Pb, and Cu, with a distance factor of 20 being the boundary. The Ag–Pb–Zn group is the medium-temperature elemental assemblage, and Sn (and Zn/Pb) represent the high-temperature elemental assemblage. The enrichment conditions for Cu may be different as compared with the other ore-forming elements.

### 4.2. Factor Analysis

We tried several preliminary tests, and finally chose the method of "principal components" to extract the factors and calculated the factor score of each group of data. Employed Kaiser's rule [42], taking eigenvalues of more than 1 as account, two factors with cumulative loading variance of 60.215%, could explain more than 60% of the total variance of the exploration data set. Other factors (eigenvalues < 1) were not considered, because they explain very low percentages of variance (Figure 7a, Table 3).

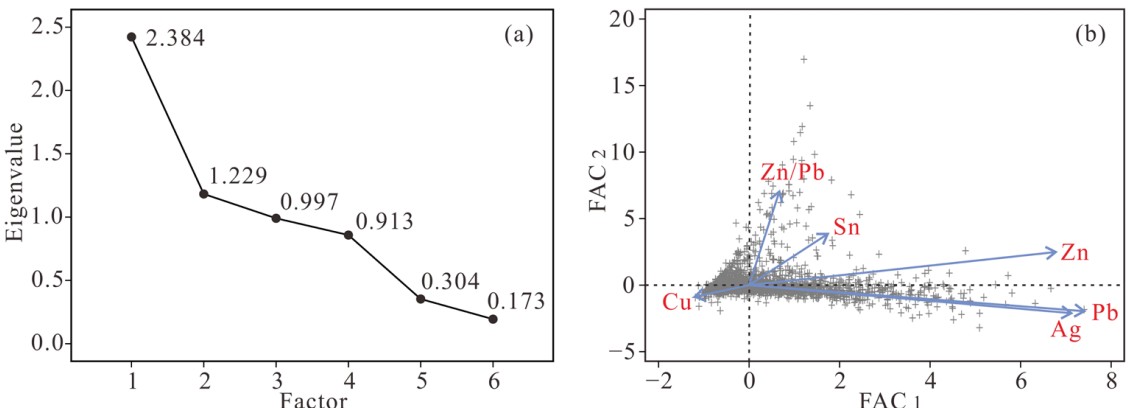

**Figure 7.** Factor analysis results. (**a**) Scree plot; (**b**) Factor score plot of samples ($FAC_1$–$FAC_2$) and component arrows of elements.

**Table 3.** Component matrix and total variance of factor analysis.

| Elements | Factor 1 | Factor 2 |
|:---:|:---:|:---:|
| Ag | **0.879** | −0.261 |
| Pb | **0.914** | −0.242 |
| Zn | **0.836** | 0.307 |
| Sn | 0.217 | **0.480** |
| Cu | −0.151 | −0.110 |
| Zn/Pb | 0.083 | **0.876** |
| Percent of variance (%) | 39.726 | 20.489 |
| Cumulative loading (%) | 39.726 | 60.215 |

Extraction method: principal component analysis.

Factor 1 accounts for 39.726% of the total variance, with significant positive loadings of Ag, Pb and Zn in the initial component matrix of the analysis (Figure 7b, Table 3). The loadings are mainly medium-temperature elements, and Factor 1 is the medium-temperature factor.

Factor 2 accounts for 20.489% of the total variance, with positive loadings of Sn and Zn/Pb in the initial component matrix of the analysis (Figure 7b, Table 3). Factor 2 is the high-temperature factor.

Factor 1 and Factor 2 scores of each sample, denoted as $FAC_1$ and $FAC_2$, are used in contours and trend surface analysis.

### 4.3. Analysis of the Semivariogram of Zn/Pb Values

As the Zn/Pb ratios can be used to trace the migration direction of metallogenic fluid, it was analyzed as a semivariogram. Variogram modelling uses samples with equal lengths. The samples with non-zero Zn and Pb were extracted and recombined according to the basic sample length of 1 m to obtain 15,294 samples. The basic statistical results of Zn/Pb and lg(Zn/Pb) values are shown in Table 4. Histograms of the frequency distributions are shown in Figure 8. The lg(Zn/Pb) values are similar to a symmetrical distribution, and thus the lg(Zn/Pb) values were used for the semivariogram analysis.

**Table 4.** Statistical characteristics of Zn/Pb and lg(Zn/Pb).

| | Number of Samples | Minimum | Maximum | Mean | Variance | Standard Deviation | Coefficient of Variation |
|---|---|---|---|---|---|---|---|
| Zn/Pb | 15,294 | 0.001 | 356.980 | 4.281 | 217.585 | 14.751 | 3.446 |
| lg(Zn/Pb) | 15,294 | −2.000 | 2.553 | 0.247 | 0.194 | 0.440 | 1.786 |

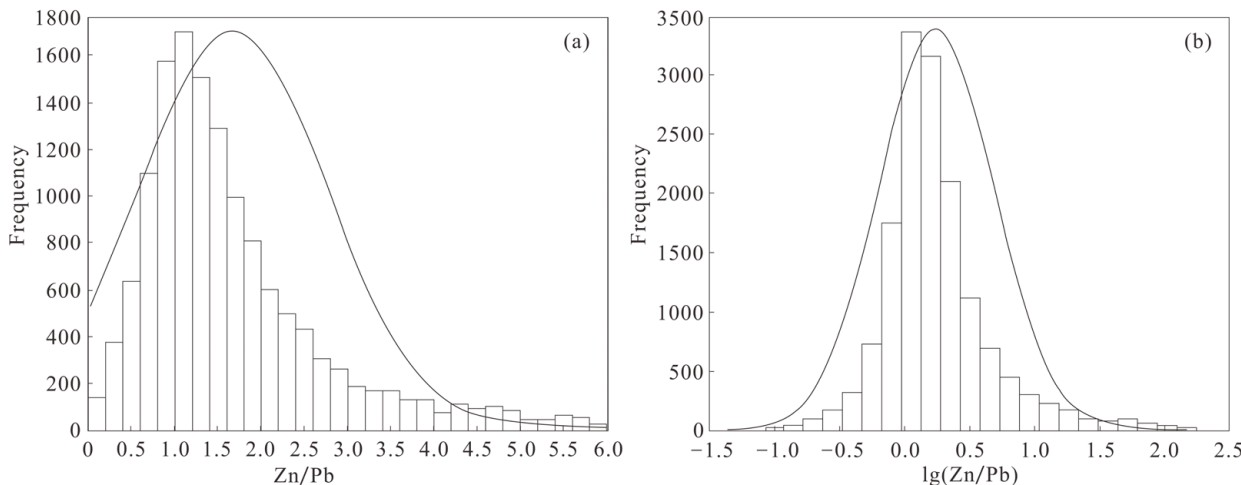

**Figure 8.** Frequency histograms of Zn/Pb (**a**) and lg(Zn/Pb) values (**b**).

The distribution of the omnidirectional semivariogram can be used to determine the main directions of the spatial correlations and relative magnitude of the correlation distances. The omnidirectional semivariogram map (Figure 9a) shows that the values are clearly anisotropic, with a good mineralization continuity in the 100° and 10° directions, with a ratio of approximately 2:1 between the two ranges. The two directions are also the strike and the dip opposite of the ore body, respectively.

The nugget/sill values in the 100° and 10° directions are similar, with values of 0.736 and 0.744, respectively, and the lg(Zn/Pb) values are moderately spatially dependent in both directions (Figure 9b,c and Table 5). The ranges in the 100° and 10° directions are 91.061 and 50.480 m, respectively, and the lg(Zn/Pb) values have a longer spatial correlation range in the 100° direction.

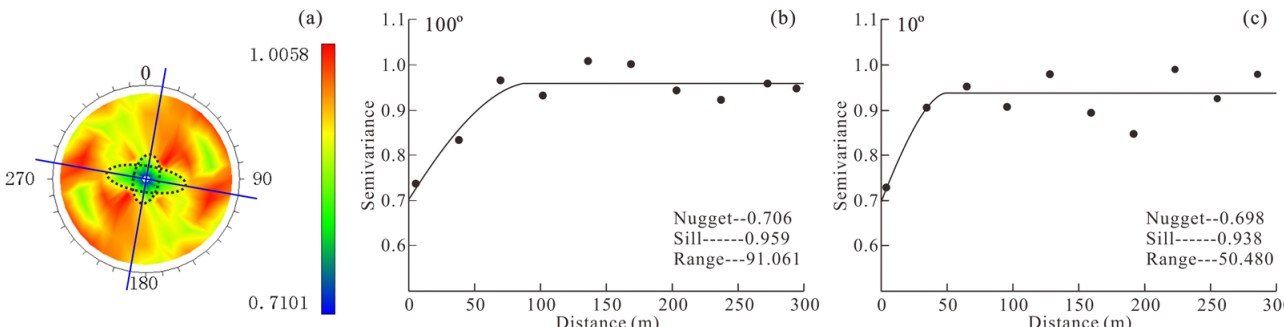

**Figure 9.** Semivariogram map of lg(Zn/Pb) values. (**a**) Omnidirectional semivariogram distribution map; (**b**) 100° direction semivariogram; (**c**) 10° direction semivariogram.

**Table 5.** Results of semivariogram analysis.

| Direction | Nugget | Sill | Range | Nugget/Sill | ME | RMSE | MSE | RMSS |
|-----------|--------|------|-------|-------------|-----|------|-----|------|
| 100° | 0.706 | 0.959 | 91.061 | 0.736 | 0.000 | 0.296 | 0.000 | 1.009 |
| 10° | 0.698 | 0.938 | 50.480 | 0.744 | 0.000 | 0.297 | 0.001 | 1.007 |

Cross-validation of the results of the semivariogram analysis using the Kriging evaluation (Table 5) shows that the mean error (ME) is close to 0, the RMSE is 0.296 and 0.297, the mean standard error (MSE; 0.000 and 0.001) is small and the RMSS is 1.009 and 1.007 and close to 1 in both the 100° and 10° directions, respectively. These parameters verify that the results of the variance function analysis are effective.

### 4.4. Ore Grade Model and Distribution of Mineralization

Using a $1 \times 1 \times 1$ m cube as the basic unit block, the internal space of the ore body can be represented by 41,441,000 unit blocks to model the ore grade. Based on the Ag, Pb, Zn, Sn and Cu grades obtained from the prospecting data, $FAC_1$ and $FAC_2$ scores from factor analysis, the grades of each unit block were estimated by ordinary kriging to obtain an ore grade model for the ore body. A horizontal projection of the ore grade was constructed based on the element grades, Zn/Pb values and $FAC_1$ and $FAC_2$ scores of each unit block (Figure 10).

High Ag, Pb and Zn grades in the ore body commonly overlap (Figure 10a–c), with high-value zones in the intervals 45–70 and 109–130 along the exploration line, and good continuity of high Ag and Pb values in the 100° direction and for Zn in the 100° and 10° directions. The high Sn grades are concentrated between the intervals 45–60 and 109–130 (Figure 10d). The high Cu grades are mainly distributed throughout the interval 45–70 (Figure 10e). The Zn/Pb values are generally >0.7 (Figure 10f), and the high values commonly overlap those of Sn. The high Zn/Pb values are concentrated in two intervals (45–60 and 109–130), with good continuity along the strike and in the dip direction of the ore bodies. The concentration of the medium-temperature factor score $FAC_1$ is the combination of Ag, Pb, and Zn, and the high-temperature factor score $FAC_2$ commonly overlaps those of the Sn and Zn/Pb values.

### 4.5. Mineralization Trend Surface Analysis

Given the gentle dip of the ore body, the grade was projected onto a horizontal plane and a two-dimensional trend surface analysis was carried out. A total of three orders of trend surface analysis was performed and the first-order trend surface analysis was selected by comparing the P, $R^2$, RMSE, CoeffVar and Durbin–Watson D values for each analysis result (Table 6). A first-order mineralization trend plane is plotted in Figure 11.

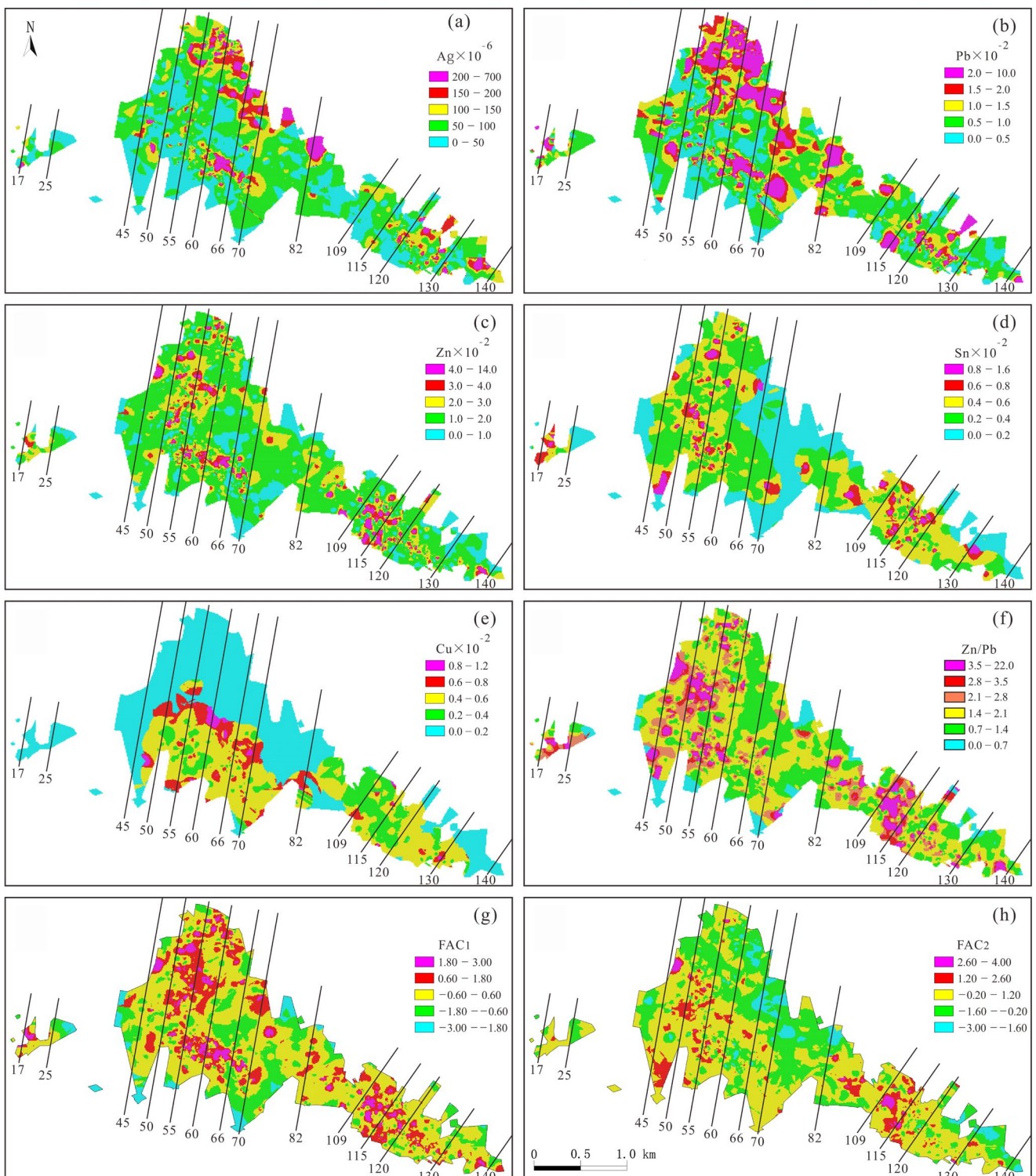

**Figure 10.** Horizontal projection of the ore grades and distribution of mineralization: (**a**) Ag; (**b**) Pb; (**c**) Zn; (**d**) Sn; (**e**) Cu; (**f**) Zn/Pb; (**g**) $FAC_1$ = medium-temperature factor score; (**h**) $FAC_2$ = high-temperature factor score.

**Table 6.** General ANOVA for significance of increasing degree of trend surfaces from first to third degree.

|  | Degree | P | $R^2$ | RMSE | CoeffVar | Durbin–WatsonD |
|---|---|---|---|---|---|---|
| Ag | First | <0.0001 | 0.00200 | 59.32018 | 62.48622 | 0.00000 |
| | Second | 0.0012 | 0.00299 | 59.29733 | 62.46215 | 0.01000 |
| | Third | 0.0012 | 0.00299 | 59.29733 | 62.46215 | 0.01000 |
| Pb | First | 0.0002 | 0.00236 | 0.89989 | 81.76063 | 0.00360 |
| | Second | 0.0007 | 0.00241 | 0.89995 | 81.76569 | 0.00371 |
| | Third | 0.0007 | 0.00241 | 0.89995 | 81.76568 | 0.00371 |
| Zn | First | 0.0067 | 0.00005 | 1.75923 | 85.90814 | 0.00005 |
| | Second | 0.0082 | 0.00006 | 1.75934 | 85.91320 | 0.00011 |
| | Third | 0.0082 | 0.00006 | 1.75934 | 85.91320 | 0.00011 |
| Sn | First | <0.0001 | 0.00711 | 0.31191 | 62.15284 | 0.01035 |
| | Second | 0.0002 | 0.00793 | 0.31185 | 62.14134 | 0.01200 |
| | Third | 0.0002 | 0.00793 | 0.31185 | 62.14133 | 0.01200 |
| Cu | First | <0.0001 | 0.02900 | 0.19243 | 37.84532 | 0.04613 |
| | Second | <0.0001 | 0.02900 | 0.19243 | 37.84532 | 0.04613 |
| | Third | <0.0001 | 0.03065 | 0.19232 | 37.82383 | 0.04795 |
| Zn/Pb | First | 0.0178 | 0.00038 | 2.19412 | 105.67910 | 1.11281 |
| | Second | 0.0234 | 0.00061 | 2.19410 | 105.67820 | 1.11324 |
| | Third | 0.0235 | 0.00061 | 2.19410 | 105.67820 | 1.11324 |
| FAC$_1$ | First | <0.0001 | 0.04344 | 1.00988 | −2008.32700 | 1.66388 |
| | Second | <0.0001 | 0.04753 | 0.99009 | −1968.96900 | 1.69927 |
| | Third | <0.0001 | 0.04751 | 0.99009 | −1968.98000 | 1.69927 |
| FAC$_2$ | First | <0.0001 | 0.09477 | 0.93944 | 24.688.91000 | 1.72459 |
| | Second | <0.0001 | 0.09537 | 0.90976 | 23.908.80000 | 1.73611 |
| | Third | <0.0001 | 0.09538 | 0.90976 | 23.908.77000 | 1.73611 |

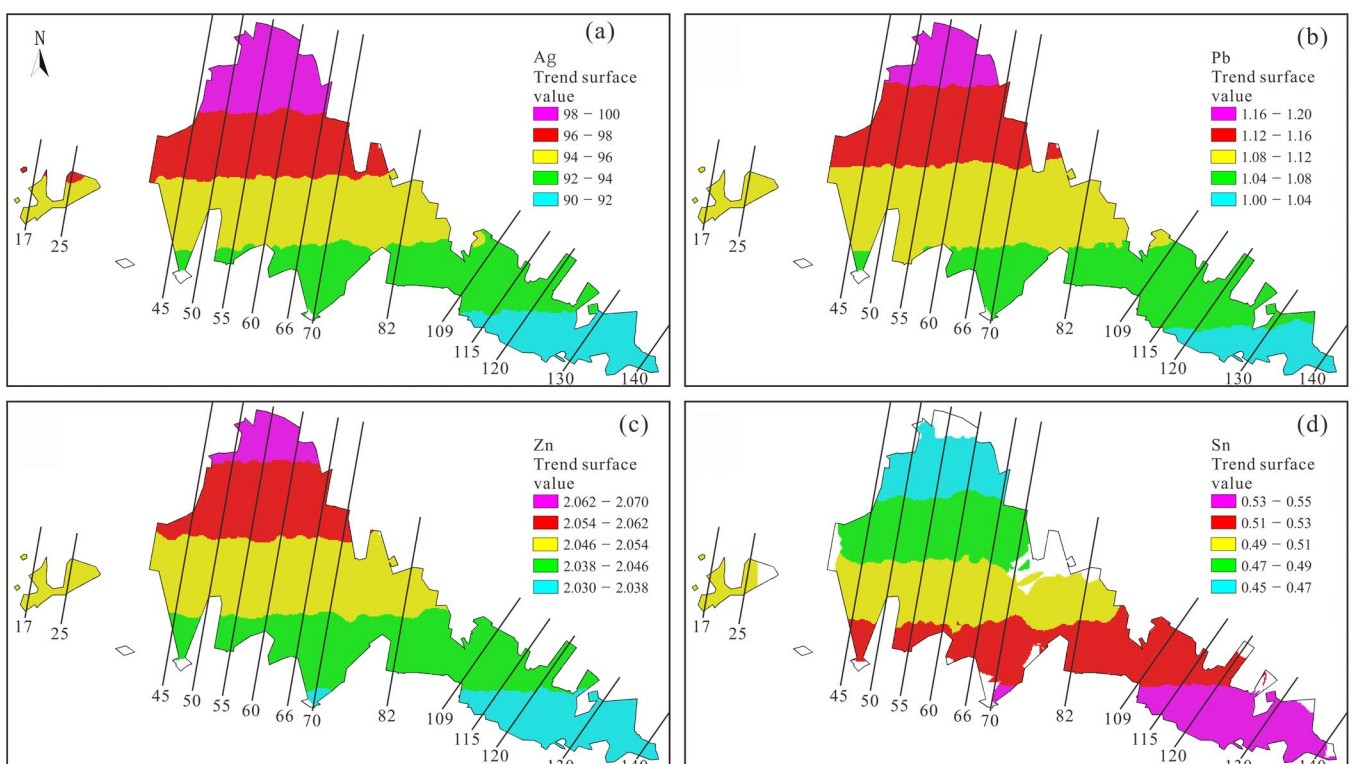

**Figure 11.** *Cont.*

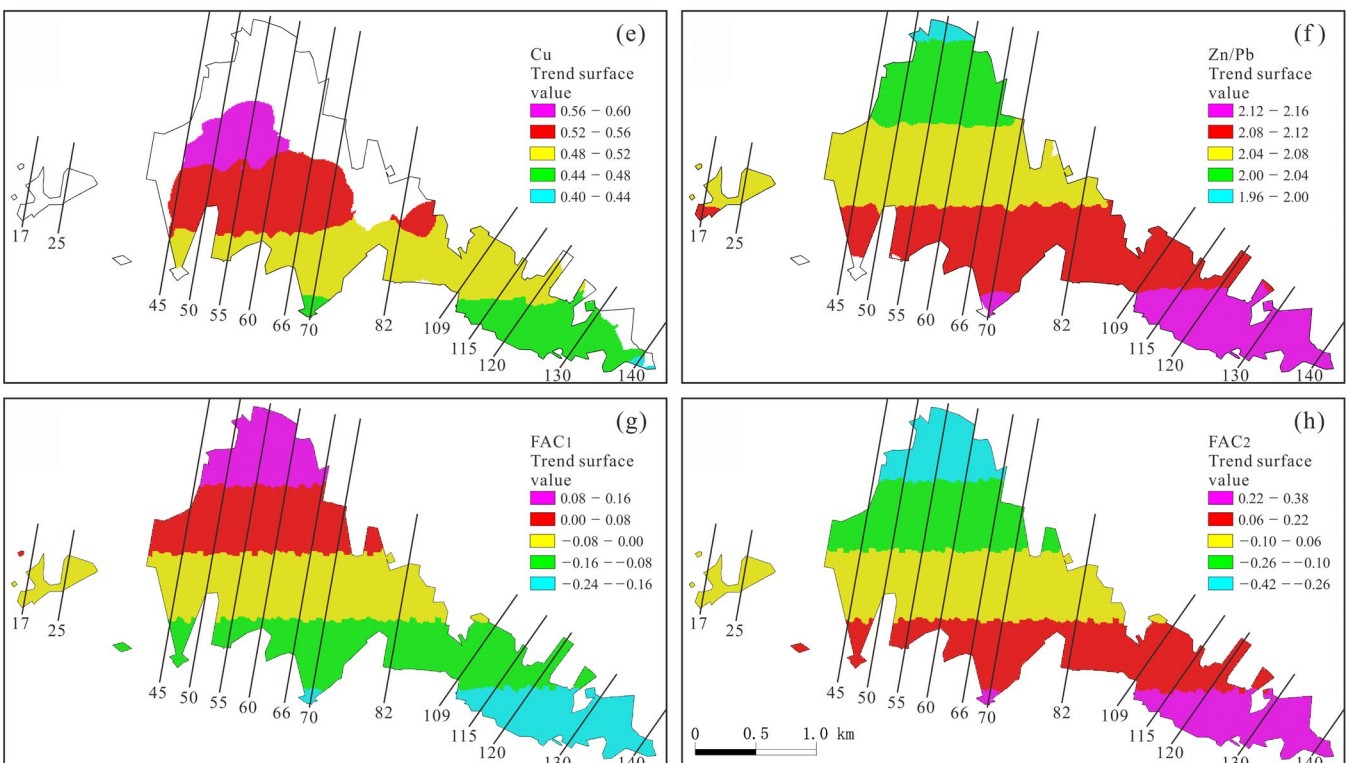

**Figure 11.** Horizontal projection of the ore body and the first-order trend surface of the mineralization: (**a**) Ag; (**b**) Pb; (**c**) Zn; (**d**) Sn; (**e**) Cu; (**f**) Zn/Pb; (**g**) FAC$_1$ = medium-temperature factor score; (**h**) FAC$_2$ = high-temperature factor score.

The first-order trend plane shows the grade trend contours for each element are distributed in a nearly E–W direction, with Ag, Pb, Zn, Cu and FAC$_1$ being weak in the south and strong in the north, and Sn, Zn/Pb values and FAC$_2$ being high in the south and low in the north. In general, the mineralization is enriched in Sn (high-temperature mineralization) in the south, Cu (medium-temperature mineralization) in the center, and Ag–Pb–Zn (medium-temperature mineralization) in the north.

## 5. Discussion

The correlation, cluster and factor analysis showed that Ag–Pb–Zn are close, this group of elements represents the medium-temperature ore-forming elements. The Ag-bearing minerals in this deposit are argentite, stephanite, pyrargyrite and jamesonite. It is assumed that these Ag-rich sulphides were syngenetic with galena and sphalerite, and that Ag–Pb–Zn precipitated from ore-forming fluid during the mesothermal stage to form polymetallic sulphide ores.

Tin is a high-temperature ore-forming element, and higher Zn/Pb values represent higher ore-forming temperatures. As such, there is a good correlation between Sn and Zn/Pb values. Tin-rich minerals in the deposit include cassiterite, stannite and franckeite. Automorphic cassiterite is enclosed by other sulphides forming a mosaic texture (Figure 3h,i), which indicates that this cassiterite crystallized prior to the other minerals as the temperature of the ore-forming fluid decreased.

The Zn/Pb values are indicative of polymetallic mineralization in the mining area. The Zn/Pb values are almost lg-normally distributed and the omnidirectional semivariogram for lg(Zn/Pb) values show that the two main extension directions are along the strike and in the dip direction of the ore body. These two directions were chosen to separately carry out the semivariogram calculations, and the nugget/sill values are similar in the 100° and 10° directions, with both showing a moderate spatial dependence. The semivariogram ranges for these two directions are 91.061 and 50.480 m, respectively.

The Ag and Pb mineralization spatially overlap, with continuous mineralization being mainly in the 100° direction and strong in the north. The Zn mineralization is continuous in the 100° and 10° directions, and Cu mineralization is continuous in the 100° direction, with a high mineralization intensity in the 50–70 interval. The Sn mineralization and high Zn/Pb values spatially overlap, with continuous mineralization mainly in the 10° direction and stronger mineralization mainly in the 45–60 and 109–120 intervals.

Trend surface analysis shows that the mineralization intensity of Ag, Pb, Zn, Cu and medium-temperature factor score $FAC_1$ are weak in the south and strong in the north, while the mineralization intensity of Sn and high temperature factor score $FAC_2$ are strong in the south and weak in the north. We hypothesize that the high-Zn/Pb zone in the south represents the source of the ore-forming fluid, which precipitated the high-temperature element Sn in the south, Cu in the center and the medium-temperature elements Ag–Pb–Zn in the north, as the ore-forming fluid migrated from south to north.

## 6. Conclusions

We used statistical methods to analyze the distribution of Ag, Pb, Zn, Sn and Cu in the Bainiuchang Ag polymetallic deposit. The elements Ag–Pb–Zn represent medium-temperature mineralization, Sn represents high-temperature mineralization and Cu was independently mineralized. The Zn/Pb values are indicative of mineralization, and the high-Zn/Pb zone was the source of the ore-forming fluid. Semivariogram analysis showed that Zn/Pb values are moderately spatially dependent, with good continuity along the strike and dip direction of the ore body, with variation ranges of 91.061 and 50.480 m, respectively. The elemental distribution in a plan view of the ore body shows overlap in Ag–Pb–Zn–$FAC_1$ and Sn–Zn/Pb–$FAC_2$ mineralization, which is highly correlated and indicates the reliability of the statistical results.

Based on the pattern of mineralization, semivariogram analysis and mineralization trend analysis, the source of the ore-forming fluid in the mining area is in the south. During migration of the ore-forming fluid from south to north, high-temperature Sn, medium-temperature Cu, and medium-temperature Ag–Pb–Zn mineralization occurred, which formed the elemental and mineralization enrichment patterns. There are concealed granites in the southeast of the mining area, and further research is needed to investigate whether these provided the ore-forming fluid.

**Author Contributions:** Methodology, F.J., C.Y., X.L. and Z.S.; formal analysis, F.J., C.Y., G.Z., M.X. and X.L.; investigation, F.J., C.Y., X.L., W.D. and J.D.; resources, G.Z., M.X., X.L., W.D. and J.D.; data curation, F.J., C.Y., X.L. and Z.S.; writing—original draft preparation, F.J.; funding acquisition, X.L. All authors have read and agreed to the published version of the manuscript.

**Funding:** The research was funded by the Yunnan Major Scientific and Technological Projects (Grant 202202AG050006).

**Data Availability Statement:** Not applicable.

**Acknowledgments:** We would like to thank Yongqing Chen of the China University of Geosciences, Beijing, for his valuable suggestions.

**Conflicts of Interest:** The authors declare no conflict of interest.

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
