# Peer review of "Mineralization Regularities of the Bainiuchang Ag Polymetallic Deposit in Yunnan Province, China"

_minerals, doi:10.3390/min13030418_

Round 1
Reviewer 1 Report (Previous Reviewer 1)
Hello to dear authors
Many thanks for your detailed revisions and corrections.
English proofreading by a native English editor is usually required. Despite the major changes you made, the English language still has main flaws. I could comment on the English language of your manuscript so far. I leave the decision about the English to the dear editor.
Line 73 and other lines : "Three-dimensional spatial positioning" is not a professional expression. You can say : "3D Spatial Modelling" or "3D Modelling". Please check this point throughout the text.
My decision is " Accept after minor revision ".
Best regards
Author Response
Thank you very much for your valuable comments. Details of the revisions to the manuscript are attached.

Reviewer 2 Report (Previous Reviewer 3)
This manuscript entitled “Mineralization regularities of the Bainiuchang Ag Polymetallic deposit in Yunnan Province, China” in current version are well organized and well written. The paper has potential to be accepted after a minor revision. One little question is about the source of ore-forming fluids, which is listed in the appendix PDF.

Author Response
Thank you very much for your valuable comments. We carefully revised and supplemented each of the suggested parts of the paper. Details of the manuscript modification are as follows:
Point : The source is unclear. please revise. The ore-forming fluids were derived from meteoric water? or magmatic fluids? or formation water?
Response : Nowadays,the source of the ore-forming fluids in the Bainiuchang deposit hasn't a clear conclusion. Previous studies have mainly focused on magmatic-hydrothermal, submarine exhalative sedimentary , or a combination of both.
Reviewer 3 Report (Previous Reviewer 2)
Dear Authors,
Thanks for submitting your manuscript to Minerals.
The below fields must be improved:
1. replacing Pearson correlation with a non-parametric criterion for valid correlation analysis.
2. using another method for clustering rather than linear correlation coefficient.
3. clarification about how the composite samples are calculated from data of analyzing samples with non-equal lengths. it is critical for variogram calculations.
4. it will we useful to summarize primary stats of raw variables in a table.
5. it is mandatory to report the results of comparison tests in the trend analysis.
6. review the Fig. 8. the log diagram does not show the relevant min-max values of the raw data. probably, the natural logarithm (Ln) is calculated.
More details can be found in the attached pdf file.

Author Response
Thank you very much for your valuable comments. Details of the revisions to the manuscript are attached.

Reviewer 4 Report (New Reviewer)
This is a comprehensive and useful investigation.
However, there are some remarks:
(1) English must be improved.
(2) The Conclusion(s) absolutely do not contain estimations necessary for industrial exploration.
(3) I believe that the Introduction should include brief information about similar polymetallic deposits in other countries.
For instance, the polymetallic deposit Filizchay on the southern slope of the Greater Caucasus (Azerbaijan):
Alizadeh, A.A., Guliyev, I.S., Kadirov, F.A., and Eppelbaum, L.V., 2017. Geosciences in Azerbaijan. Volume II: Economic Minerals and Applied Geophysics. Springer, Heidelberg – N.Y., 340 p.
Author Response
Thank you very much for your valuable comments. We carefully revised and supplemented each of the suggested parts of the paper. Details of the manuscript modification are as follows:
Point 1: English must be improved.
Response 1: Thank you for your valuable and thoughtful comments. We have carefully checked and improved the English writing in the revised manuscript.
Point 2: The Conclusion(s) absolutely do not contain estimations necessary for industrial exploration.
Response 2: This part has been removed.
Point 3: I believe that the Introduction should include brief information about similar polymetallic deposits in other countries. For instance, the polymetallic deposit Filizchay on the southern slope of the Greater Caucasus (Azerbaijan): Alizadeh, A.A., Guliyev, I.S., Kadirov, F.A., and Eppelbaum, L.V., 2017. Geosciences in Azerbaijan. Volume II: Economic Minerals and Applied Geophysics. Springer, Heidelberg – N.Y., 340 p.
Response 3: This part has been added.
This manuscript is a resubmission of an earlier submission. The following is a list of the peer review reports and author responses from that submission.
Round 1
Reviewer 1 Report
Hello to dear authors
Thank you for your efforts in expanding mineral exploration techniques.
Your manuscript is generally suitable for publication in Minerals, but there are some points about scientific structure and presentation, please read them and revise the manuscript carefully:
Title
(1) The title is not appropriate. The “title” should be descriptive, direct, accurate, appropriate, interesting, concise, precise, unique, and should not be misleading. For example : Geochemical investigations to identify the mineralization controlling factors using analytical methods, ....deposit... .
Abstract :
(2) You need to rewrite abstract section entirely. It is not acceptable now. please consider these parts in writing : Background, Methods, Results, and Conclusions respectively. The first sentence is your best effort to attract the reader by stating the main field.
Introduction :
(3) Lines 35 to 50 (First paragraph of intro.) are not suitable. please write about the significance and generality of your research. please rewrite the first paragraph. Please use all of these references to show the importance of using analytical methods to achieve the best results:
Fusion of Remote Sensing, Magnetometric, and Geological Data to Identify Polymetallic Mineral Potential Zones in Chakchak Region, Yazd, Iran
Neuro-Fuzzy-AHP (NFAHP) Technique for Copper Exploration Using Advanced Spaceborne Thermal Emission and Reflection Radiometer (ASTER) and Geological Datasets in the Sahlabad Mining Area, East Iran
Fusion of Lineament Factor (LF) Map Analysis and Multifractal Technique for Massive Sulfide Copper Exploration: The Sahlabad Area, East Iran
(4) Line 54: You only mention the tonnage of economic elements in the Bainiuchang deposit. Please also mention the considered threshold concentration (For each element).
(5) Line 56: You have already said the full name of the deposit (Bainiuchang Ag polymetallic deposit), please avoid unnecessary repetition, it will make the text boring.
(6) Line 56: What do you mean by the term "ore-forming materials" ? Use specialized terms correctly throughout the text.
(7) Lines 68 to 74: You must explain the methods used in the research and the reason for using them. What methods were used before? What do you mean by traditional methods? What are the advantages and disadvantages of applied methods compared to traditional methods? please rewrite this paragraph carefully.
(8) Lines 74 to 77: You must state the objectives of the research in this section. Please write the main objectives of your research clearly and sequentially in this paragraph.
Geological Background
(9) Please prepare a figure of structural zones of China and locate the study deposit on it. International persons will be read your paper.
(10) Please mention the type of mineralization in the study deposit and explain its general characteristics.
Data compilation and statistical methods :
(11) Please change the section name to "Raw Data and Methodology".
(12) Lines 122 to 130 : please summarize the exploratory operations (Drill holes, tunnels and etc) in a Table.
(13) Line 133 : As I found the 3D model of ore body was prepared by you. you must move it the Results section. you should talk about the data used here.
(14) Please summarize the sub-section of "3.3. Geological significance of Zn/Pb values " as a paragraph in the introduction.
(15) You should prepare a comprehensive flowchart of your technical flow and methodology. please put it at the start of the methodology section.
Results
(16) Line 229 : Good analysis ! please write about the "Ternary diagrams " in the methodology section.
(17) Figure 9 : please use a high quality image. Also add the units of the values to the figure.
(18) Why you don't use all of the drill holes? you mentioned, there are 408 drill holes, before. A limited number of boreholes are shown in Figure 9. If it is a section, specify the location of the section.
The English language of your manuscript is very poor. This article must be read by an editor and all spelling, composition and grammar errors must be corrected. One of the repeated flaws in the text is the word Mineralization / Mineralisation , which is repeated throughout the article.
Best Regards
Author Response
Thank you very much for your valuable comments. We carefully revised and supplemented the article according to suggestions. The quality of the revised manuscript has been greatly improved, and our writing level and professional ability have also been improved during the revision process. Details of the manuscript modification are attached.

Reviewer 2 Report
Comments, suggestions and corrections can be found in the reviewed version of the manuscript. There are critical points about the manuscript:
1. There is very few information about the deposit type. We know that elemental zoning patterns are different for different types of the same metals.
2. Statistical methods are not appropriate due to the nature of data you used. Suggestions are in the pdf file.
3. When we speak about the correlations, cluster analysis and geostatistical analysis of element grades in mining samples, we must consider their compositional nature. Compositional data analysis (CoDa) is not mentioned in the manuscript. You may neglect to use CoDa methods, but this will decrease the work quality and validity.
in general, a combination of CoDa and Universal kriging (or better, geostatistical simulation) is suggested.
cm: please use foxit reader to open the reviewed pdf document.

Author Response

(The authors gave the same response as above.)

Reviewer 3 Report
This manuscript has potential to be accepted after a major revision. All comments and suggestions are listed in the appendix PDF for your consideration.

Author Response

(The authors gave the same response as above.)
